# Microstructure and Hydrophobicity of the External Surface of a Sonoran Desert Beetle

**DOI:** 10.3390/biomimetics7020038

**Published:** 2022-03-31

**Authors:** Luis E. Tellechea-Robles, Rodrigo Méndez-Alonzo, Francisco E. Molina-Freaner, Amir Maldonado

**Affiliations:** 1Posgrado en Ciencia de Materiales, Universidad de Sonora, Hermosillo 83000, Mexico; luistellechea91@gmail.com; 2Departamento de Biología de la Conservación, Centro de Investigación Científica y Educación Superior, Carretera Ensenada-Tijuana 3918, Ensenada 22860, Mexico; mendezal@cicese.mx; 3Departamento de Ecología de la Biodiversidad, Instituto de Ecología, Universidad Nacional Autónoma de México, Hermosillo 83000, Mexico; freaner@unam.mx; 4Departamento de Física, Universidad de Sonora, Hermosillo 83000, Mexico

**Keywords:** hydrophobicity, micropatterned surfaces, beetles

## Abstract

We have studied the external surface (elytra) of the Sonoran Desert beetle (*Eleodes eschscholtzii*). Our aim was to assess whether this species has similar traits to some beetles from the Namibian Desert that are known to have hierarchical micropatterns that allow for water harvesting. We have conducted scanning electron microscopy (SEM) and apparent contact angle experiments on specimens collected at two sampling sites with different ambient humidity. The results show that the beetle’s external surface microstructure is composed of a compact array of polygons with randomly scattered protuberances. The density of the polygons in the microstructure is different for individuals collected in different sites: the polygon array is denser in the more humid site and less dense in the drier site. The measured contact angles also depend on the sampling site. For individuals collected in the drier site, the average apparent contact angle is 70°, whereas for the more humid site, the average apparent contact angle is 92°. FT-IR experiments are consistent with the presence of hydrophobic wax compounds in the studied surfaces. Our investigation opens new questions that are currently being addressed by experiments that are underway. For instance, it would be interesting to know whether the observed nanopatterns could be used in biomimetic devices for water harvesting purposes.

## 1. Introduction

Living organisms adapt to their environments in order to take advantage or protect themselves from weather conditions, escape from predators, and find food and water sources. Nature uses material properties for these purposes. This is particularly true in desert or semidesert regions where water is scarce and plants and animals need to optimize water-capture and -retention strategies. One can take inspiration from plants or animals to prepare surfaces suitable for water harvesting applications [1,2].

Several beetle species from drylands have water harvesting capabilities through air moisture condensation in their elytra surfaces (the hard, external crust protecting their wings). Some Namib Desert beetle species are known for their ability to collect water in this way [3,4,5]. In this case, the water harvesting mechanism involves an array of micro-patterned motifs in the insect’s external surface; the patterned areas promote water condensation, so beetles can harvest fog in micro-droplets for water consumption [6]. An important component of this mechanism used to collect fog consists of adopting a head-standing position opposite to the air currents, and with an angle large enough to allow droplets to form and roll down to the beetle’s mouths [7,8] (Appendix A).

Previous reports in the literature have studied the water collection mechanism that desert beetles develop through their most external surfaces. Namib Desert beetles have the ability to condense small water droplets for consumption. However, the physical mechanism has not been fully understood despite its biomimetic potential.

On a microscopic scale, the external layer of these beetles consists of an almost random array of hydrophilic bumps; their sides and troughs are covered by a microstructured, hydrophobic wax coating. Water collection starts in the peaks, and once water strikes the hydrophobic slopes, it is collected by rolling down the tilted beetle’s surface. Several groups have taken inspiration from the patterned structures observed in nature in order to build novel biomimetic materials [6,9,10]. Some interesting examples include the following: The biomimetic potential of the beetle’s surface structures has been determined by creating a bidimensional array of glass spheres on a wax-coated microscope slide; this simple device was able to collect water with a better efficiency than a hydrophobic surface [10]. Hydrophilic patterns on superhydrophobic surfaces with water harvesting characteristics similar to that of the Namib Desert beetle have been created using polyelectrolyte-layered films [6]. In an interesting report, a superwetting material (a stainless-steel, mesh filter) has been modified according to the beetle’s back structure. The created, biomimetic filter proved to efficiently extract water from water-in-oil emulsions. In fact, the oil fraction in the emulsion permeated through the mesh pores and purified up to 99.95 wt%, demonstrating a great application potential for water-in-oil emulsion separation and oil purification [9].

The mentioned Namibian beetles belong to the Tenebrionidae family, whose members are native to warm places, such as the deserts of Africa and North America (including the Sonoran Desert). In drylands, species from this group, particularly those in the genus *Stenocara*, use moisture condensation to collect drinking water. The mechanism that these insects seem to employ to harvest water droplets from moisture in the air is based on a highly structured, external surface. This surface is hydrophobic but is furnished with hydrophilic protuberances. Water condensation starts on the protuberances. When large enough droplets are formed, they fall into the hydrophobic region, where, with the right tilting, they slide and move to the insect’s mouth [7].

Other species from the Tenebrionidae family are worth studying in terms of microstructure and hydrophobicity, as they display similar behaviors to those of Namib Desert beetles. In particular, it is interesting to investigate whether North American beetles also display microstructures similar to those that confer water-collection abilities to the African beetles.

In this context, we report an investigation, mainly by means of scanning electron microscopy experiments and apparent contact angle measurements, of the microstructure and hydrophobicity of the external surface (elytra) of *Eleodes eschscholtzii*, a species of Tenebrionidae from the Sonoran Desert. Our aim is to assess whether this species has similar morphological traits to those of the beetles from the Namibian Desert. For this, we have collected specimens in two sites clearly differentiated by yearlong, atmospheric humidity. One of the chosen sites (a coastal one) is subject to more frequent fog as compared to an inland location. One would thus expect that, if surface microstructure and hydrophobicity are relevant for fog-collection, they should differ between the chosen sites. Finally, we explored the chemical composition of the elytra to test for the presence of epicuticular waxes that could promote hydrophobicity. Our research may shed light on how the studied beetles adapt to the climate and on their capabilities of harvesting water. At the same time, this work may provide new inspiration for novel water-harvesting, biomimetic technologies.

The paper is divided as follows: In Section 2, we provide the experimental details of our investigation. In Section 3, we present and discuss the main results. Finally, we draw some conclusions.

## 2. Materials and Methods

### 2.1. Sampling Sites and Capture Traps

We selected two sites that vary in climate, particularly in atmospheric moisture, in the Sonoran Desert to collect desert beetles (Figure 1). One site is located on the southeastern border of the city of Hermosillo, Sonora (29°1′18.6″ N, 110°57′10.1″ W). The other site is located 100 km away in a coastal area in Kino Bay (28°48′51″ N, 111°55′38.1″ W). Hereafter, the sampling sites will be called HMO (inland) and KB (coastal). Even though the sites are located around 100 km apart, there are several climatological differences between them. Meteorological records (1951–2000) from HMO indicate that mean annual rainfall (MAR) is 305 mm and mean annual temperature (MAT) is 24.3 °C, but in KB, MAR is 135.6 mm and MAT is 20.5 °C [11]. More important for our study, there are clear differences in atmospheric relative moisture between both sampling sites. Records of relative humidity from Weather Atlas show that HMO is a drier site during the year, whereas KB is a more humid site during the year (Climate-Data) (see Appendix A), most likely due to a marine influence associated with its location on the coast of the Gulf of California. In fact, the relative humidity is roughly 20% higher in KB than in HMO (see Appendix A).

Once the sites were selected, we installed 25 pitfall traps along a 100 m transect, using groups of 5 traps per site (Appendix A) every 20 m. Transects were identical in each locality and oriented in a northwesterly (NW) direction. We used antifreeze fluid (ethylene glycol) in each trap to preserve fallen insects. We kept pitfall traps in each site for 7 days before collecting the captured insects for analysis.

### 2.2. Studied Specimens

Among the insects collected in the pitfall traps, we selected nine, individual desert beetles from each sampling site. The set of 18 collected specimens varied in length from 1.5 to 3 cm. They were carefully dissected in order to extract elytra to characterize its surface microstructure and hydrophobic properties (Appendix A). These individuals were identified as members of the species *Eleodes eschscholtzii* (Figure 2A).

*Eleodes* (Tenebrionidae, Eschscholtz, 1829) is the name of a group of North American beetles which includes ca. 200 species [12]. *Eleodes eschscholtzii* (Solier, 1848) is distributed in the North American drylands from the SW United States to NW Mexico, including Baja California [13].

### 2.3. Hydrophobicity Quantification Using Water Apparent Contact Angle

Apparent contact angle (CA) [14,15] measurement is a standard procedure for quantifying the degree of hydrophobicity of any surface. It is conventionally assumed that a surface can range from hydrophilic if the apparent contact angle ranges from 0 to less than 90°, neutral if close to 90°, and hydrophobic if measurements pass 90° to 180° [16]. This measurement is conventionally performed with optical tensiometers.

Samples were transported under refrigeration less than 2 weeks after collection to the Conservation Biology Department in CICESE, Ensenada, Mexico, where CA was measured for all of the collected specimens from the KB sampling site, but for only 5 insects from HMO, due to difficulties in obtaining flat sections of elytra. We used an optical Tensiometer–Goniometer (Theta Lite, Attension, Biolin Scientific, Västra Sweden). A mean number of 64 photographs, taken at a speed of 150 fps, were produced for each sample, and using Attension software, contact angles (CA) were averaged for both left and right angles.

Note that a full wetting characterization would be interesting, including contact angle hysteresis. We reserved those experiments for a future investigation.

### 2.4. Scanning Electron Microscopy

To obtain amplified images of the elytral surface, we used an Olympus SZ51 stereoscopic microscope at a magnification of 5×. We also used a scanning electron microscope (SEM: JEOL, JSM-5410LV, Tokyo, Japan) at the Polymers and Materials Department (DIPM), Universidad de Sonora. Each sample was visualized at three amplifications: 200×, 1000×, and 2000× using secondary electrons to improve resolution and enhance the detection of morphologies. For all samples, we used a voltage between 15 and 20 kV.

For purposes of comparison and correlation between apparent contact angle measurements and microstructures observed in the SEM images, samples were separated into three groups: low hydrophobicity samples, neutral hydrophobicity samples, and high hydrophobicity samples [16].

### 2.5. Fourier Transform Infrared Spectroscopy

We used a PerkinElmer FT-IR Spectrometer (PIKE Technologies GladiATR) from the Department of Polymers and Materials at the Universidad de Sonora. Previous studies characterizing biophysical properties of cuticular lipids in insects have used FT-IR analysis [17]. It has been reported that the spectra display a small peak in the 2800–3000 cm^−1^ range, which corresponds to the absorption range of hydrocarbon groups like methylene (-CH2-) and methyl (-CH3). These groups are characteristic of elytral waxes.

For the chemical composition analysis, we used three samples: the most hydrophobic sample (individual 5 KB, CA: 104°), the most hydrophilic sample (individual 1, HMO, CA: 52°) and an intermediate sample (individual 9 from KB), with a contact angle (CA) of 75°. Measurements were performed in the spectral range of 400–4000 cm^−1^, using a total of 16 scans per sample.

## 3. Results

In order to obtain an idea of the microstructure of the external surface of the studied beetles, we took optical (stereoscopical) and scanning electron microscopy images. In Figure 2B,C, we show a section of the elytra of a specimen collected in HMO. Even with no magnification (B), a channel-like pattern can be observed. At 8× (C), this pattern is clearer. It consists of a parallel array of small microchannels. The periodicity of this array is of the order of half a millimeter.

In Figure 3, we present representative scanning electron microscopy images of the elytra surface of a Sonoran Desert beetle. Two characteristic features are clearly observed at this microscopic scale. First, one can see that the surface is completely covered by an array of polygons composed primarily by hexagons and pentagons; this array is reminiscent of a bee’s honeycomb. The polygons have an effective diameter slightly superior to 10 μm. The second feature distinguished in Figure 3 is the existence of larger, circular dots scattered and randomly distributed on the surface. The diameter of these dots is of the order of 40 μm, and they are separated at a distance of the order of 150–200 μm. From the center of the spots, short filament-like structures seem to protrude. Note that both the honeycomb-like texture and the array of protuberances have been identified in other beetle elytra: *Stenocara gracilipes* [8], *Physasterna cribripes* [18], and *Stenocara eburnea* [19]. In some of these species, this microstructure plays a role in their water-harvesting capabilities. It is interesting to compare the geometrical dimensions of the Sonoran Desert beetle microstructures to those reported in the literature. The bumps and polygons displayed in the elytra of *S. eburnea* have similar dimensions to those in the specimens studied in this work. In other species, the size of the polygons is quite similar, but the diameter of the bumps is larger, of the order of 200–300 μm [10,18].

A closer examination of the polygon array reveals interesting characteristics (Figure 4). First, the patterns are composed of both hexagons and pentagons. This could be an indication that the surfaces bear some curvature since a flat structure can be completely filled with hexagons, whereas a curved surface requires both hexagons and pentagons or other geometric figures. Second, the polygon density is different for specimens collected at different sampling sites. In fact, the polygon array is denser for beetles from the more humid site (KB) than for those from the drier one (HMO). From micrographs like those in Figure 4, we measured the number of polygons per unit area and the polygon area in the elytra. The average polygon area is different for both sampling sites: 166 μm^2^ for HMO and 130 μm^2^ for KB. At this stage, it is not clear whether this difference is related to some physical or physiological property. Additional SEM images are presented in Appendix A.

In order to have an idea of the hydrophobicity of the Sonoran Desert beetles’ external surface, we have measured the water apparent contact angle (CA) of specimens collected in the described sites. Apparent contact angle values for every individual captured at both sites are shown in Appendix A, where the mean value and the standard deviation are also reported. It is interesting to note that apparent contact angles differ between specimens from the two sites. CA values obtained for beetles from HMO are somehow lower than those measured for beetles from KB. The former ranged from 52° to 90°, whereas the latter ranged from 75° to 104°. The average apparent contact angle (and standard deviations) is 70° ± 18 for specimens from HMO and 92 ± 9° for those from KB.

The measured apparent contact angles indicate that the external surface of specimens from HMO is hydrophilic on average, while the elytra of specimens from KB is slightly hydrophobic. It is interesting to note that HMO is an inland site, with higher summer precipitation but lower relative humidity (RH) through the year, whereas KB is a coastal site with lower precipitation, but higher RH. In contrast with relative humidity, the annual rainfall rate is very low in both sites and is not a relevant factor to consider in the environmental conditions. Thus, the experimental results suggest that, although beetles from both sites are members of the same species, differences in the atmospheric moisture of their habitat reflect on their physical properties, such as the water contact angle, of their external surfaces. In the studied case, we observed that desert beetles living in habitats with higher atmospheric moisture have more hydrophobic elytra than those living in habitats with lower atmospheric moisture. In fact, a plausible natural solution to the water-harvesting necessities of these desert beetles is the modulation of the external contact angle in function of differences in air moisture.

We attempted to relate the morphological surface patterns observed in the samples to the measured apparent contact angles. For this, we determined for each sample the average area per polygon and the average polygon density (per unit area). The number of polygons was estimated in a 60 μm by 60 μm square in the SEM micrographs (Appendix A). The results are presented in Appendix A, along with the respective contact angle. We can observe that the samples differ not only in the contact angle value, but also in the polygon area (and thus, density). For instance, the more hydrophilic sample (CA = 52°) displays polygons of a mean area equal to 308.7 µm^2^, whereas the more hydrophobic sample (CA = 104°) has polygons of a smaller area: 119.2 µm^2^. In Figure 5, we present boxplots comparing the data for individuals collected in both sampling sites.

Note that smaller polygons (or higher polygon density in the honeycomb-like microstructure) should contribute to a higher hydrophobicity. In fact, high polygon densities should enhance the formation of air pockets in the gaps between polygons, thus promoting larger apparent contact angles [14,15] (see discussion and Figure 8). In order to have more insight into this idea, we plotted the contact angle value as a function of the number of polygons per unit square (60 by 60 μm^2^), as shown in Figure 6. We can appreciate that, as a general trend, the apparent contact angles increase as the number of polygons increases. However, if one looks for a quantitative correlation between the data with a linear regression analysis, the coefficient of determination is rather low (R^2^ = 0.38). This fact indicates that the measured contact angles are poorly correlated with the average polygon number per unit area. This is probably an indication that other factors, such as the elytra curvature, may also contribute to the hydrophobicity of the surfaces. Further experiments are necessary in order to clarify this issue.

The external surface of beetles (elytra or cuticle) is a protective layer. It prevents water from evaporating and shields the insect body from mechanical stresses. It is known that it contains proteins, chitin, and aliphatic compounds [20,21,22]. In order to have an idea of the chemical composition of the elytra surface of the studied Sonoran beetles, we performed Fourier Transform Infrared spectroscopy (FT-IR) experiments. The results are presented in Figure 7 and Appendix A. Several absorption peaks and bands are observed. They are due to the compounds present in the insect cuticle. To single out just some of the features in the spectra, we can signal the peaks observed in the region 3000–2850 cm^−1^, which correspond to aliphatic compounds such as those present in waxes. There are also several absorption peaks that correspond to proteins or chitin. The FT-IR results show that the external surface of the Sonoran beetles is composed of the same elements as other, similar species. Further experiments are necessary in order to determine more precisely the composition and the spatial arrangement in the composite material forming the elytra. The FT-IR spectra are consistent with the presence of hydrophobic compounds in the analyzed surfaces. Note that aliphatic compounds, as well as chitin, may contribute to the hydrophobicity of the beetle’s elytra. However, the distribution of these molecules along the cuticle may not be homogeneous [23,24,25,26].

## 4. Discussion

We have shown so far that the external surface of the Sonoran Desert beetle *E. eschscholtzii* is covered with a microscopic array of polygons with scattered protuberances of larger dimensions; the microstructures are somehow similar to those reported for other beetles, namely those from the Namibian Desert. Our experiments also provide evidence that the elytra´s chemical composition includes proteins, chitin, and aliphatic compounds. Moreover, individuals from sampling sites with different mean ambient humidity display different hydrophobic behaviors. It would be interesting to perform more experiments and analysis in order to determine whether these differences are related to the water-harvesting capabilities of these insects. From the point of view of materials science, it would also be interesting to study the water-capture behavior of surfaces with compositions and structures similar to those reported in this study.

Note that similarly patterned structures found in other desert beetles allow for water harvesting. Indeed, hydrophilic protuberances function as water-collecting regions; air moisture condensates in these protuberances, whereas the flat, polygon-covered surfaces function as nucleation zones, where hydrophobic compounds allow droplets to nucleate and be transported by successively larger channels. As a result, the insects are able to capture water from the only available source during periods of several months in an extremely dry environment. In this context, water condensation via intercaled bumps within a layer of different hydrophobicity is probably a water-collection mechanism of the studied beetles.

The main challenge in the biomimetic materials field is to take inspiration from the structures observed in nature and to create artificial models with efficient operations. Water-harvesting efficiency tests have been performed with some species from the Namibian Desert (*O. unguicularis*, *O laeviceps*, *S. gracilipes,* and *P. cribripes*). In those experiments, the beetles’ heads were located down, at an angle of 23° inside a fog chamber, imitating the natural tilting angle. In these conditions, the volume of water collected after two hours was of the order of 0.15 mL [8]. Several groups have proposed biomimetic surfaces inspired by the beetles’ elytra [6,27,28]. In one of these studies [27], a hydrophilic polymer array was deposited on a superhydrophobic background; with these structured surfaces, the water harvesting yield was 2–3 g/cm^2^ after a two-hour collection time. In another interesting report [28], several materials including polytetrafluoroethylene, aluminum, and carbon nanotubes were used as substrates to create patterned surfaces; with 1.8 cm by 1.8 cm samples, it was possible to harvest around 0.8 g of water in a 5 min collection time.

Note that two wetting regimes of rough surfaces are possible (Figure 8). In the Wenzel state, the liquid penetrates the grooves of the solid surface. In the Cassie–Baxter, or heterogeneous state, air pockets remain trapped in the grooves. The latter case results in a higher apparent contact angle. Given the patterned surfaces observed in Figure 3 and Figure 4, it is plausible that the wetting state of the Sonoran Desert beetles is of the Cassie–Baxter type. However, more experiments are needed in order to elucidate this issue.

From a biological perspective, it is also interesting to try to understand the possible way in which evolution has shaped the observed elytra morphology. In order to comprehend whether this morphology is indeed an adaptation to capture fog, it would be necessary to perform a comparative analysis with individuals from different beetle species. To the best of our knowledge, there has been no thorough search for analogous microstructures across commonly originating species to study fog-harvesting properties. This issue should be considered a priority for biomimetic surveys. In fact, if these traits are preserved across the evolutionary history of any specific group of organisms, the implication would be that natural selection is operating to optimize a particular fog-harvesting structure, which should become an obvious target for replication. In addition, further experimentation should test whether the contact angle and the density of polygons in the two-dimensional arrays may vary during the development of these organisms under varying levels of relative humidity.

## 5. Conclusions

We have studied the hydrophobicity and microstructure of the external surface (elytra) of Sonoran Desert beetles (*E. eschscholtzii*) from two sampling sites with different ambient humidity. Our results show that the elytra of the studied American species display geometrical traits similar to those of African beetle species (from the Namibian Desert) known for water-harvesting capabilities.

In our case, a scanning electron microscopy experiment revealed that the surfaces are covered by an array of hexagons and pentagons placed in such a way as to fill the bidimensional space. The density of the polygon array depends on the sampling site. The array is denser in specimens from the more humid site, whereas it is less dense for those from the drier site. Moreover, the surface hydrophobicity also depends on the humidity of the sampling site: the surfaces of specimens from the more humid site are hydrophobic on average, whereas those of specimens from the drier site are slightly hydrophilic on average.

Additional experiments are underway in order to determine whether the observed traits are related to the water-harvesting capabilities of these insects. If this is the case, it would also be interesting to develop biomimetic structures inspired by the observed surfaces.

## Figures and Tables

**Figure 1 biomimetics-07-00038-f001:**
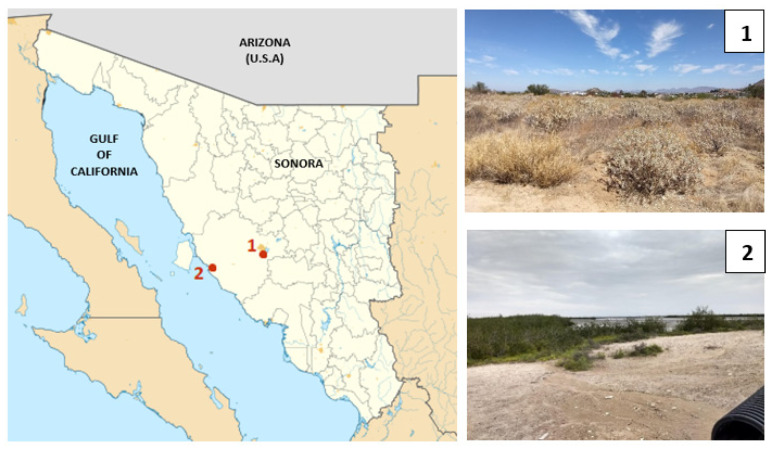
Sampling sites in the Sonoran Desert. Red dots show the location of each site: (**1**) Hermosillo (HMO) and (**2**) Kino Bay (KB).

**Figure 2 biomimetics-07-00038-f002:**
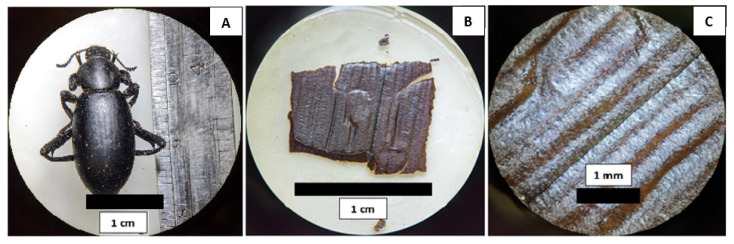
(**A**) Dorsal view of one, individual *E. eschscholtzii* collected from HMO. Channel-patterned structures of the external surface observed with a stereoscopic microscope at no magnification (**B**) and 8× (**C**).

**Figure 3 biomimetics-07-00038-f003:**
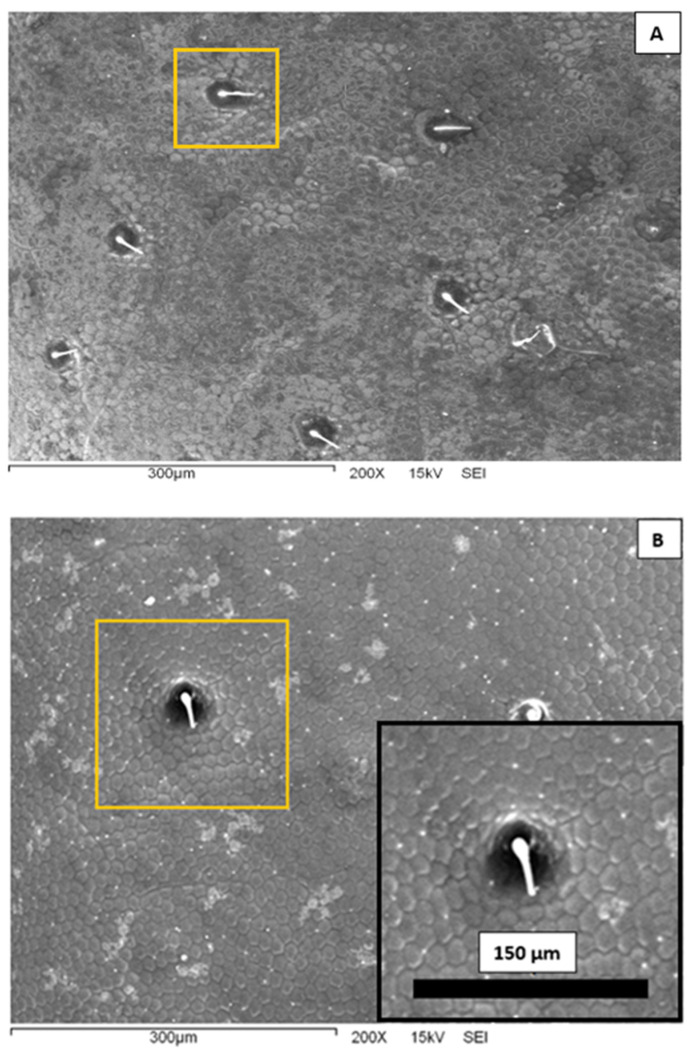
Scanning electron microscopy images of the external surface of a Sonoran Desert beetle at 200× magnification. (**A**) The background is covered with an array of polygons. There are also protuberances in a random distribution. One is highlighted in a yellow square. (**B**) A closer view of the surface. The inset shows a further-amplified protuberance.

**Figure 4 biomimetics-07-00038-f004:**
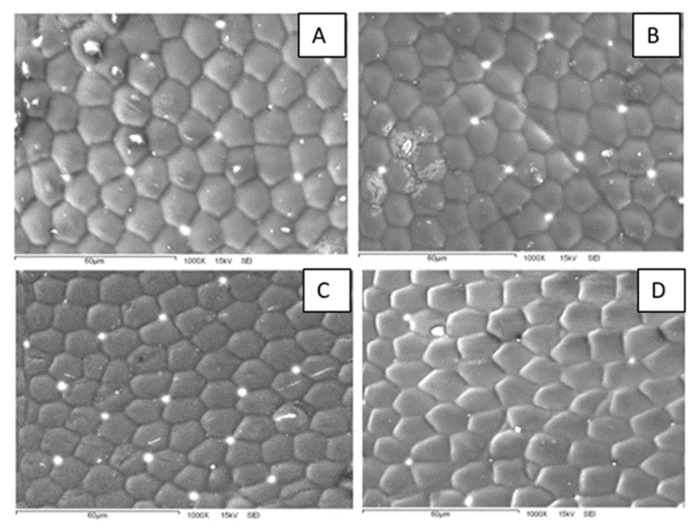
SEM micrographs of the polygon arrays in the background surfaces of *E. eschscholtzii* individuals. (**A**,**B**) Specimens from the HMO collecting site; (**C**,**D**) Specimens from KB.

**Figure 5 biomimetics-07-00038-f005:**
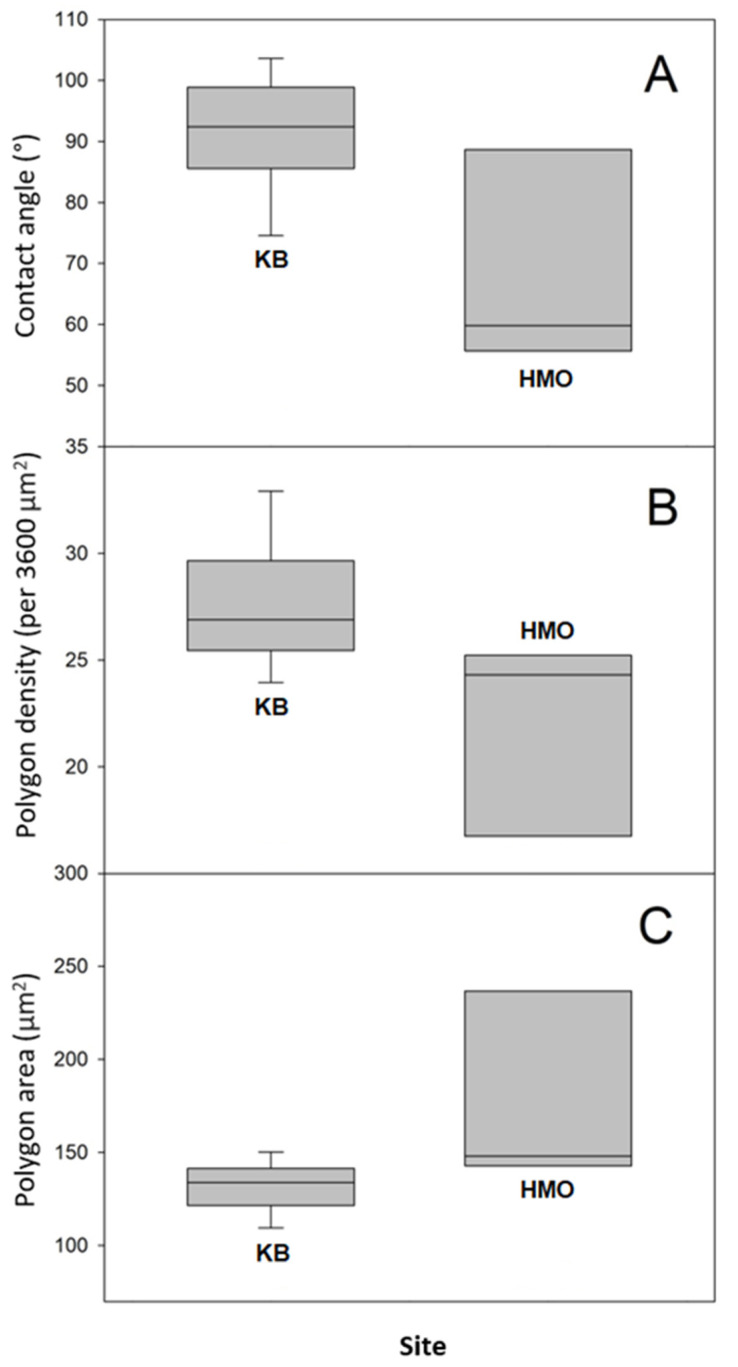
Boxplots of apparent contact angle and microscopic measurements performed to contrast the hydrophobicity and microstructures of Sonoran Desert beetle specimens collected in both sampling sites (**A**), contact angles; (**B**) polygon densities; and (**C**) average polygon areas.

**Figure 6 biomimetics-07-00038-f006:**
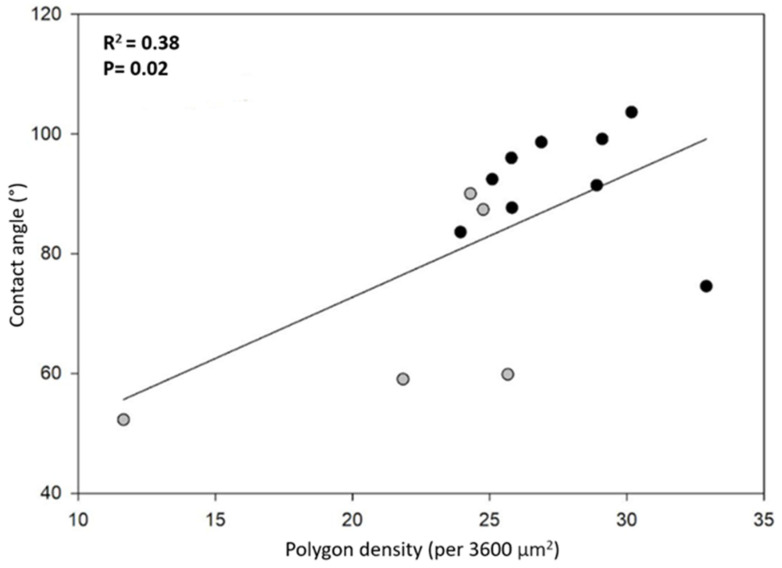
Correlation between contact angle and polygon density per area (3600 µm^2^): HMO (**gray dots**) and KB (**black dots**).

**Figure 7 biomimetics-07-00038-f007:**
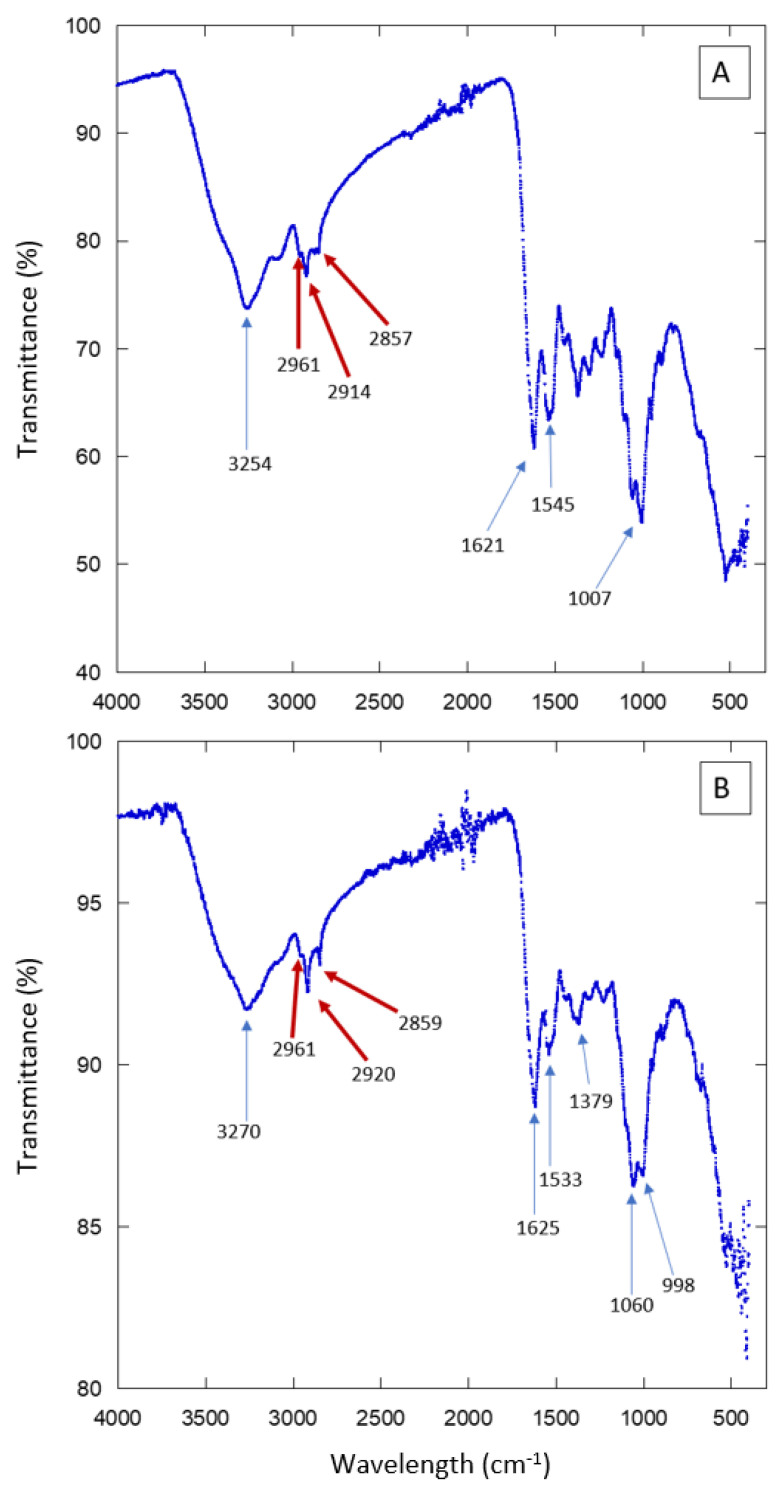
FT-IR scans of elytra from individuals collected in HMO (**A**) and KB (**B**).

**Figure 8 biomimetics-07-00038-f008:**
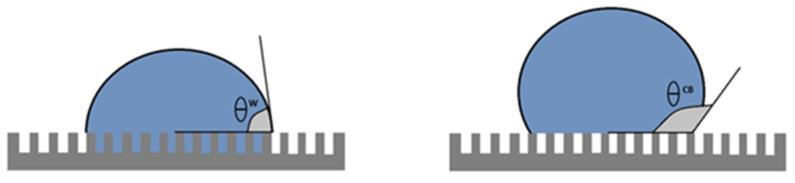
Wenzel-type wetting (**left**); and Cassie–Baxter-type wetting (**right**). In the Cassie–Baxter wetting behavior, water droplets are separated from the solid surface due to small air pockets trapped between the grooves.

## Data Availability

Not applicable.

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
