# Peer review of "Microstructure and Hydrophobicity of the External Surface of a Sonoran Desert Beetle"

_biomimetics, 2022, doi:10.3390/biomimetics7020038_

Round 1

Reviewer 1 Report

In this study, Tellechea-Robles et al. studied the hydrophobicity and microstructure of the external surface (elytra) of a Sonoran Desert beetle collected from two sampling sites with different ambient humidity. It is interesting that the microstructure (the polygon array) and the related wettability is different on the beetle external surface (elytra) from different sampling sites. For the sample from the more humid sampling site, the polygon array is denser and the surface is hydrophobic, whereas the polygon array is less dense and the surface is slightly hydrophilic for the drier site. It is concluded that the elytra of the studied American species display similar geometrical traits as an African beetle species (from the Namibian Desert) known for water harvesting capabilities. However, the manuscript cannot be considered for acceptance by this journal in its present form. The proper revision is necessary for its resubmission. Below are comments that may help improve it:

  1. To our knowledge, the desert beetles have been extensively studied in biomimetic field, especially their water collection ability reported widely (Nature 2001, 414, 33; Frontiers in Zoology 2010, 7, 23; ACS Nano, 2017, 11, 760; Nano Lett. 2006, 6, 1213; Bioinspir. Biomim. 2014, 9, 031002; J. Bionic Eng. 2009, 6, 63). However, the authors only simply introduced few reports and discussed the mechanism for water collection originated from surface structures. Compared to those mentioned studies which elucidate the relation between the wettability and the beetle surface structures for guiding the bioinspired fabrication of functional structures and devices, what is the breakthrough of this study? I strongly suggest that the authors should summarize and compare the advances on the wettability of desert beetles systematically to highlight the novelty of this study in the Introduction.
  2. This study only explored the structure-mediated wettability of the beetles in nature reported everywhere, which is not mentioned the biomimetic structure and performance from these beetles. As for the focus of this journal Biomimetics on biomimicry and bionics, this seems to be far from journal standard. Thus, the biomimetic experimental results should be given to verify the discovery from other beetles.
  3. In the Materials and Method section, the authors referred to “there are clear differences in atmospheric relative moisture between both sampling sites”. It is easy to measure or obtain the atmospheric relative moisture at both sampling sites. To better get know of the difference more intuitively by other researchers, the authors must provide the specific value of atmospheric relative moisture at sampling sites like the value of annual rainfall provided. In addition, the authors used a voltage between 15 and 20 kV for the SEM characterization of all samples. Did this voltage damage the biological surface because 5 kV is usually used?
  4. The authors captured more specimens for observation and characterization. But when calculating the polygon density on the beetle external surface, the polygon areas are only calculated. What is the number of polygons per unit area? How much sample number to use for calculation? Does the sample number influence the polygon density?
  5. This manuscript stated that the elytra of the studied American species display similar geometrical traits as an African beetle species (from the Namibian Desert) known for water harvesting capabilities. By checking the reported Namibian beetle, we found the bumps/protuberances on the external surface have different sizes for both beetles. So, the description seems to be improper.
  6. The authors only studied the structure and wettability of the beetles by means of scanning electron microscopy experiments and contact angle measurements. But for the beetles, the water harvesting capability is crucial to explain the structure and wettability differences the authors found because these structures and functions were evolved to adapt to their living environment.

Author Response

We have carefully read the reviewers´ suggestions. We have modified the manuscript accordingly. The revisions are highlighted in red in the new manuscript. We thank the reviewer for the insightful comments.

1.- We have added the new references the reviewer suggests. In addition we have summarized these and other works in the introduction. We think that with this we highlight the novelty of our study: we investigate a new beetle, from another continent (America) as compared to the previous studies (African beetle).

2.- We have added a comment on the discussion section describing the performance of some of the previously studied beetles. We also add comments on some biomimetic investigations and give the yield of some of the described biomimetic surfaces. Relevant references are added.

3.- We added a lines in the materials and methods sections explaining the difference in relative humidity between the sampling sites. We also direct the reader to the plot in the Supplementary materials. The used voltage in the SEM experiments did not damage the biological samples. In fact, our experiments were highly reproducible: the same structures were observed in all the samples. In addition, since the surfaces were dry we did not expect any damage to the samples.

4.- We added some lines in the Results section in order to clarify these experimental details. Sample number did not influence the polygon density. The required number of polygons per unit area is given in the text, in figure 5 and in Table 1 of the supplementary materials all the data are presented. We analyzed all the collected samples.

5.- We have added some comments in the results section in order to compare the geometrical dimensions of the studied micropatterned surfaces with those reported in the literature for the Namibian beetle. We think the manuscript is better now.

6.- We agree with the reviewer. However, the water harvesting behaviour and yield of our beetles is out of the scope of this paper where we just wanted to characterize the microstructure. Further experiments are underway in order to be able to answer this issue.

Reviewer 2 Report

The paper presents interesting and novel results. It is publishable after the minor revision.

Remarks:

  1. The notion of the apparent contact angle should be used for the characterization of the reported surfaces, see: Drelich L. et al. Contact angles and wettability: towards common and accurate terminology, Surface Innovations, 5 (1), 2017, pp. 3-8; Bormashenko Ed. Physics of Wetting. Phenomena and Applications of Fluids on Surfaces, de Gruyter, Germany, Berlin, 2017 . The use of the accurate scientific wording is extremely important.
  2. Not only apparent contact angles, but also the contact angle hysteresis should be discussed under the revision (see the aforementioned references, please).
  3. It is absolutely impossible to establish apparent contact angles with an accuracy of four significant figures, as it is reported in the manuscript. The correct rounding of the results should be carried out.
  4. What kind of wetting takes place in the reported experiments? Is it the Cassie- or Wenzel-like wetting regime?

Author Response

We have carefully read all the reviewer's suggestions. When necessary, we have modified the manuscript in order to take them into account.

Specific answers:

1.- We thank the reviewer. We use now the accurate wording in all the manuscript (apparent contact angle). We also added the two suggested references.

2.- We added a line on this issue in the Materials and methods section. We agree that it will be interesting to study contact angle hysteresis but this kind of experiments were not the focus of the present paper. We let them for a future work. We agree that it will be very interesting to fully characterize the wetting behaviour.

3.- We thank the reviewer. We agree. We have modified all the apparent contact angle values. We have rounded all the values to present integer numbers.

4.- Since the micropatterned surface has irregularities or grooves it is plausible that the wetting state is that corresponding to a Cassie-Baxter type. We discuss this issue in the manuscript and add a new figure.

Round 2

Reviewer 1 Report

Although Tellechea-Robles et al. revised the manuscript through adding proper discussions, the supplement experiments are still lack. As for biomimetic filed, the inherent performance from biological organisms is important for biomimetics, but the surpassing nature is more critical. Thus, I think the demonstration of the water harvesting capability is necessary. After all, the authors referred to water harvesting devices in key words. Compared with the reported beetles, what is the advantages? If there is no advantage, what is the meaningless of this study? If it only displays the relationship between hydrophobic and surface structure of the biological organism extensively reported, I think it is lack of novelty. In conclusion, as for the focus of this journal Biomimetics on biomimicry and bionics, the biomimetic experimental results should be given to verify the novelty.

The authors addressed that “To the best of our knowledge, there is no report in the literature on the hydrophobic and microstructural properties of the species studied in this work.” I think this statement is improper because the hydrophobic and microstructural properties widely exist in natural species (plants, insects, animals, fish, etc.) which reported in many publications. If this is the novelty, many studies will occur through study single species. I think the aim to study the species is better to find the universality and the underlying mechanism, thereby inspiring us to construct high-performance materials or devices. It is not only the simple investigation of the species to discovery a phenomenon.

Besides, as for a systematic study, the enough experimental data is necessary. The authors highlight the hydrophobic property but not offering a full wetting characterization like contact angle hysteresis. It simply said that “We let those experiments for a future investigation”.  I strongly suggest to supplement the related experiment because of adequate revision time.

Author Response

We thank the reviewer for his/her comments. However, we do not fully agree. The aim of the paper was just to characterize the surface of the studied beetle. In fact, we selected this journal because many published papers have the same characteristics (see some references at the end of this response). In all these papers the systems are just characterized and there is no attempt to create a biomimetic device. We see our paper is of the same quality of those previously published.

In addition, we are not able to perform further experiments in a reasonable time. It is our aim for to perform these experiments for a future publication.

Anyway, we have modified the revised version. For instance, we have deleted the key words: "water harvesting devices". In addition, we have deleted the sentence: "To the best of our knowledge...".

Finally, the novelty of our work is clearly stated in the Introduction and in the Conclusions: besides studyin a new beetle species (in other continent) we have studied beetles from sampling sites with different humidity. The results show that the hydrophobicity depends on this parameter.

The following papers from Biomimetics have the same style, focus and experimental design as ours. In fact, we have selected this journal after reading some of them.

1.- Underwater Attachment of the Water-Lily Leaf Beetle Galerucella nymphaeae (Coleoptera, Chrysomelidae)

Grohmann et al, Biomimetics 2022, 7(1), 26

2.- Chitosan Extraction from Goliathus orientalis Moser, 1909: Characterization and Comparison with Commercially Available Chitosan

Fournier et al, Biomimetics 2020, 5(2), 15

3.- Variation of Goliathus orientalis (Moser, 1909) Elytra Nanostructurations and Their Impact on Wettability

Godeau et al, Biomimetics 2018, 3(2), 6

4.- Photonic Crystal Characterization of the Cuticles of Chrysina chrysargyrea and Chrysina optima Jewel Scarab Beetles

Vargas et al, Biomimetics 2018, 3(4), 30